# Chemical Sensor Nanotechnology in Pharmaceutical Drug Research

**DOI:** 10.3390/nano12152688

**Published:** 2022-08-05

**Authors:** Lebogang Thobakgale, Saturnin Ombinda-Lemboumba, Patience Mthunzi-Kufa

**Affiliations:** 1National Laser Centre, Council for Scientific and Industrial Research, P.O. Box 395, Pretoria 0001, South Africa; 2College of Agriculture, Engineering and Science, School of Chemistry and Physics, University of Kwa-Zulu Natal, University Road, Westville, Durban 3630, South Africa

**Keywords:** chemical sensors, nanomaterials, detection, pharmaceuticals, therapeutic drugs

## Abstract

The increase in demand for pharmaceutical treatments due to pandemic-related illnesses has created a need for improved quality control in drug manufacturing. Understanding the physical, biological, and chemical properties of APIs is an important area of health-related research. As such, research into enhanced chemical sensing and analysis of pharmaceutical ingredients (APIs) for drug development, delivery and monitoring has become immensely popular in the nanotechnology space. Nanomaterial-based chemical sensors have been used to detect and analyze APIs related to the treatment of various illnesses pre and post administration. Furthermore, electrical and optical techniques are often coupled with nano-chemical sensors to produce data for various applications which relate to the efficiencies of the APIs. In this review, we focus on the latest nanotechnology applied to probing the chemical and biochemical properties of pharmaceutical drugs, placing specific interest on several types of nanomaterial-based chemical sensors, their characteristics, detection methods, and applications. This study offers insight into the progress in drug development and monitoring research for designing improved quality control methods for pharmaceutical and health-related research.

## 1. Introduction

With the surge of pandemic symptoms on the rise, the need for more therapeutic medication is also increasing. Secondly, it is worth noting that with the upcoming variants of the COVID-19 virus, we should expect the demand for medication to increase further. As such, pharmaceutical companies at both the research and industrial phases will experience pressure to produce more drugs in mass without compromising quality control. In the former case, research into drug design is paramount to producing new medication with better properties, less side effects, and better efficiencies. Once successful, industrial production of new and existing medication will require extensive and thorough quality screening performed in a timely and cost-effective manner. These concerns have inspired the compilation of this review article to assess the current detection, monitoring, and analysis methods used in the pharmaceutical industry. Nanomaterials, which are compounds with size dimensions from 1–100 nanometers (nm), are playing a key role in drug research as adsorption platforms [1,2,3]. These materials carry special properties such as electrical conductivity, optical transmittance, easy surface modification, thermal conductivity, and large surface areas which are all essential requirements of a good chemical and biochemical sensor [4,5]. Furthermore, the chemical modification of nanoparticles allows the design of tailored scaffolds for the recognition and adsorption of analytes. Chemical interactions such as hydrogen bonding, electrostatics, intermolecular forces, pi–pi stacking, and ligand binding allow the sensors to collect the analyte for high sensitivity and selectivity applications [6,7,8,9,10].

Another important aspect of chemical sensors is the detection method used to produce the signal. An efficient detector must be able to recognize changes in the sensing platform upon interaction with the analyte. Therefore, careful consideration needs to be applied when choosing a sensor and detector combination [11,12]. Most research in nanomaterial-based pharmaceutical investigations employs electrical and optical detection methods for signal collection [13,14]. In the former case, electrical apparatuses such as surface-modified electrodes, conductors, and electrolytes are combined to form a circuit that can produce electrical signals such as impedance, voltage, and currency as a means of analyte detection [15,16,17]. Examples of electrical methods discussed in this review are field effect transistors (FET) and electrochemical devices [18,19,20]. Optical detection is also a key feature of this publication where photonics-based methods are explored in relation to pharmaceutical research. Organic molecules such as therapeutic drugs interact with light sources to produce signals that are used in detection [21]. Nanomaterials, especially gold and silver, offer surface plasmons that produce signal-enhancing resonance effects necessary for amplification. Localized surface plasmon resonance (LSPR) is an example of a detection method where surface plasmons are used for the detection of analytes, mostly with the aid of metallic nanoparticles (MNPs) [22,23,24,25]. Another consequence of light–matter interaction is the production of inelastic scattered radiation which corresponds to the molecular bond of the analyte. Raman spectroscopy is a method that uses this scattered radiation to characterize a variety of compounds [26,27,28]. When combined with nanomaterials, signal enhancement effects occur, which provide a detection method of high sensitivity and selectivity [29,30].

In this review, the properties of polymeric, metallic, and graphene-based nanomaterials are explored as adsorption platforms for a variety of pharmaceutical drugs for a wide range of diseases. Secondly, a discussion on electrical and optical detection methods based on parameters such as limit of detection, linear range, and sensitivity is given. Lastly, we end with future prospects and shortcomings of nanomaterial-based pharmaceutical research in relation to drug monitoring and quality control.

## 2. Nanomaterial Scaffolds and Therapeutic Drug Applications

Sensor platforms play a pivotal role in chemical sensing because they interact with the target analyte to produce a detectable change in their physical and chemical properties. Such interactions occur via chemical bonding or intermolecular forces which cause the attachment of the analyte to the nanomaterial-based sensor [31,32]. This property of a chemical sensor is crucial to the signal output quality and reliability of the detection methods and as such, in this section, we explore the different nanomaterials used to produce sensing platforms for the detection of therapeutic drugs. Figure 1 below shows several types of polymeric nanomaterials used in drug delivery applications.

The nanoparticles in the image above have been used in various applications regarding therapeutic drugs and drug delivery [34,35]. In the following sections, we focus on polymer and inorganic nanoparticles as well as graphene-based sensors to cover a broader scope of an already expanding field of nanoparticles.

### 2.1. Polymeric Nanomaterials and Their Applications

The key factors challenging the efficiency of therapeutic medicines arise from the biological barriers present in the human tissues which affect drug delivery and intracellular transport [36,37]. As a potential solution, nano-based polymers of various chemical functionalities are explored as controlled drug delivery agents for a variety of drug treatments. The interest comes from their flexibility in surface modification, biocompatibility, and loading capacity for both hydrophobic and hydrophilic drugs [38,39,40,41,42]. The current advanced methods for producing polymeric nanoparticles include sonication, emulsification, self-assembly, electro dropping, nanoprecipitation, microfluidic, and ionic gelation. The general design of polymeric nanoparticles for drug delivery applications is nanocapsules (polymeric membranes with an empty core) and nanospheres (matrix systems in solid form) [43]. Examples of nanocapsules include polymersomes, which are artificial vesicles that consist of a double membrane made from amphiphilic block copolymers. Polymersomes are known for their good stability and efficiency in drug retention during transit to the cytosol. Dendrimers are a popular example of nanocomposites that comprise hyperbranched polymers that form a three-dimensional matrix for cross-linking purposes. They have well-defined intramolecular spaces where drug encapsulation can take place [33]. The maximum amount of cargo that can be carried relies on the shape and size of the drug molecules and the number of cavities in a dendrimer. In application, the functionality or surface chemistry, as well as the size and shape, can be tailored for specific therapeutic drugs and biomolecules [44]. There are many ways that nanopolymers carry pharmaceutical drugs to the target sites. Depending on the polymer design, a drug can be encapsulated in the core of the polymer, adsorbed in the polymer matrix, or conjugated to the surface of the nanoparticle. Figure 2 below shows the synthesis and application of polymeric nanoparticles in drug delivery.

The figure shown above describes the chemical process of synthesizing PEG-Schiff-DOX conjugates for drug delivery. Anticancer drug DOX was encapsulated in a PEG-Schiff nanosphere which transports the drug into the target cell via endocytosis [45]. Using these mechanisms, a combination of modified polymers is often applied as drug carriers. For example, a pH/redox responsive stimuli sensor made from copolymers poly-ethylene-glycol (PEG) and poly-L-lysine (PLL) functionalized with platinum nanoparticles, was used to transport gemcitabine, a small molecule for the treatment of lung cancer [46]. Poly(propylene imine) (PPI) dendrimers were chemically modified with folate for targeting the anticancer drug docetaxel [47]. HIV medications efavirenz and lamivudine were also targeted using PPI dendrimers which resulted in improved drug uptake and efficacy, respectively [48,49]. In another study, an anticancer drug, oxaliplatin (IV), was cross-linked with polyethylenimine (PEI) for delivery of reactive oxygen species during chemotherapy [50]. Co-drug delivery studies have also been explored where two drugs are delivered simultaneously using polymer nanoparticles. For example, PLGA nanoparticles coated with polyvinyl alcohol (PVA) were used to transport antitumor medication paclitaxel/methotrexate complex [51]. Polymersomes of polylactic-co-glycolic acid (PLGA, inner shell) and PEG (outer shell) were loaded with anticancer therapeutics as a promising cancer drug delivery platform [52]. The survey conducted on polymeric nanoparticles shows that these nanosensors have been invaluable in pharmacology and oncology research. However, the literature has also cited a few disadvantages of polymeric nanoparticles such as toxicity and particle aggregation which weakens drug loading. As of late, only a few nanopolymer-based medicines have been FDA approved, thus this field of research is relatively new, and more experimental work and clinical trials are still required [53].

### 2.2. Metallic Nanomaterials and Their Applications

In recent decades, noble metals and some transition metals have gained interest in a variety of nano-based applications, including pharmaceutical research. The main attraction for metallic nanoparticles (MNPs) is the resonant plasmon feature that arises from the electron oscillations at the surface of metallic atoms [54]. Because of this, MNPs have played a key role in many optical detection methods as a signal enhancement platform [55]. Furthermore, the high electrical conductivity of MNPs makes them efficient electrochemical sensors in many biomedical applications. MNPs can also be tailored into various shapes and sizes as well as chemically modified with polymers and other recognition elements to suit the intended applications [56]. Nanoparticle synthesis is usually described in terms of two major groups: top down and bottom up. In the former method, bulk material is broken down into smaller fragments with sizes less than 100 nm using diverse sources of energy. Most techniques under this group are used in the fabrication of thin film sensors usually from silicon or quartz substrates [57,58]. Examples of top-down techniques include physical vapor deposition (PVD), chemical vapor deposition (CVD), and optical and electrical lithography [59,60]. For the latter case, bottom-up techniques assemble precursor reagents into nanostructures using chemical and physical methods. Examples of popular synthesis routes include self-assembly and chemical reduction [61]. Stabilizers such as sodium citrate, cellulose, thioester, dextran, gelatin, etc., are often used to prepare the MNPs and provide the starting chemical functionality for further modification [62,63]. Figure 3 below shows the basic process for the preparation of gold NPs using chemical reduction by Turkevich and Burst methods [64,65].

The above are two examples of how gold nanoparticles of various functionality are prepared using bottom-up methods. Popular shapes of MNPs produced using these methods include nanospheres, nanorods, nano-urchins, nano cubes, nanostars, and nanocages [66,67]. Imaging techniques such as atomic force microscopy (AFM), transmission electron microscopy (TEM), and scanning electron microscopy (SEM) have become the standard for characterizing MNPs for most synthesis methods [68]. Figure 4 below shows images of MNP acquired using different microscopy techniques.

Metals such as gold (Au), silver (Ag), titanium (Ti), platinum (Pt), and copper (Cu) have been extensively explored as sensing platforms for a wide range of therapeutic and biomedical experiments [71,72]. Amongst them, gold has received the most attention owing to its inertness, easy functionalization, and biocompatibility. For example, gold nanoparticles were modified with PEG to form a PEGAuNP composite for targeted drug delivery of pancreatic cancer medications doxorubicin and varlitinib [73]. AuNPs have also been explored as novel diagnostic tools for the management of melanoma cancer [74]. In other works, gold was shown to reduce toxicity, and improve immunogenetic activity and stability when used as a vaccine carrier in nanomedicine [75]. Glucose detection was demonstrated using AuNP in serum, showing detection limits in the nanomole region [76]. Metals such as silver have also played a role in the therapeutic drug research space as in the case where gallic acid-coated silver nanoparticles were used as drug carriers for doxorubicin [77]. Silver NPs are also capable of detecting uric acid, lidocaine hydrochloride, thiamine, lomefloxacin, and propafenone in urine samples with detection limits in the micromolar per liter range [78,79,80,81,82]. MNPs are showing increased value in pharmaceutical research and will grow our understanding of therapeutic agents as future work is published.

### 2.3. Graphene Based Nano-Sensors and Applications

In the past decade, the fabrication of sensors and biosensors has improved with the incorporation of graphene as a scaffolding platform. The chemical and physical properties of graphene such as high surface-to-volume ratio, electrical conductivity, optical transmittance, thermal conductivity, and high mechanical strength give the nanomaterial a considerable edge over most materials in sensing applications [83]. Furthermore, due to the sp2 hybridization of carbon bonds, graphene is easily modified with various chemical and biochemical agents making it a material with versatile applicability. Graphene is usually used in its oxygenated form, known as graphene oxide (GO), as well as reduced graphene oxide (rGO) for research work that investigates hydrophilic molecules [84]. The Figure 5 below shows the molecular shapes of graphene, GO, and rGO.

Graphene is synthesized using various methods such as the Hummers method, exfoliation or mechanical cleavage of graphite, and chemical vapor deposition [84]. To obtain GO, strong oxidizing agents such as sodium permanganate and sulphuric acid are reacted with graphite or graphene to produce hydroxyl (OH) and carboxylic acid (COOH) functional groups on the surface of the substrate. The oxygen content on GO is reduced using photochemical, microwave, and bacterial methods to produce reduced graphene oxide. GO and rGO offer extra properties suitable for chemical and biochemical sensor applications [84]. Adsorption of drug molecules, anticancer medication, polymers, proteins, genes, and other biomolecules is possible using GO and rGO because the OH and COOH groups allow easy conjugation bonding. Electrostatic interactions using the oxygen lone pair of electrons, pi–pi stacking via the aromatic rings, hydrogen bonding, and van der Waals forces are the various mechanisms by which GO and rGO adsorbs materials onto its surfaces [85]. Figure 6 below shows the adsorption of the protein BSA on rGO nanosheets.

The image above shows the synthesis of rGO from graphite oxide with the intention of delivering anti-cancer medication DOX supported by the protein BSA. The scaffolding material is then investigated using optical methods to release the DOX from the adsorption site [86]. Similar work is available in different research applications where a combination of graphene members, crosslinkers, and analyte recognition elements are combined for targeted drug delivery. For example, amino acid-functionalized iron oxide nanoparticles adsorbed on graphene sheets were used as a sensor for dopamine and ascorbic acid detection [87]. Secondly, GO functionalized with platinum nanoparticles was explored as a chemical sensor for glucose at concentrations in the millimolar range [88]. Chitosan-functionalized GO was used in the controlled release of the anti-inflammatory drug ibuprofen [89]. In the next section, more examples of graphene sensors are discussed, taking into consideration the detection methods and parameters used to classify the efficiency of a chemical sensor.

## 3. Detection Methods and Sensing Techniques

Adsorption of the analyte to the sensing material is a crucial element of a sensor, as explained in Section 2, and the chemical properties of the sensor and the analyte need to be compatible for the adsorption to occur. A second and equally important characteristic of a sensor is the detection method employed to produce a reliable and reproducible signal. There are currently various methods used for chemical sensing of pharmaceuticals, each of which has its own advantages and disadvantages. In this section, we explore electrical and optical detection methods, comparing their sensor performance using parameters such as limit of detection, linear range, and other additional information.

### 3.1. Electrical Detection Methods

The electrical potential of a chemical agent has been investigated as a means of quantifying changes in concentration or molecular bonding because of interaction with an analyte. Electrical signals such as resistance, capacitance, conductance, and impedance are the ones used to study the thermodynamic and kinetic properties of the analyte [11]. Furthermore, electron exchange between oppositely charged species in redox reactions produces a signal that can be used as a sensing mechanism. Many nanoparticle scaffolds, conjugates, and analyte recognition elements have been coupled with electrochemical detection using surface functionalized electrodes [90].

#### 3.1.1. Field Effect Transistor (FET) Based Detection Method

Field effect transistors are devices that use an electric field to control the flow of current in a semiconductor. In a typical FET system, source (S) and draining (D) electrodes are connected to a semiconductor path that is functionalized with sensing elements for high specificity and binding affinity. When target analytes are detected, a change in channel conductance is recorded and processed to acquire an electrical signal. There are two kinds of FETs based on the PN junction theory: n-type where electrons are the main charge carriers and p-type with holes as the primary charge carriers. In an n-type FET system, positively charged molecules are detected and charge carriers (electrons) accumulate on the sensing channels and increase the signal. However, when negatively charged targets are detected, the conductivity decreases caused by the depletion of the electrons. The second type is a p-type FET system, which binds to positive charges causing a decline in conductivity decline due to a reduction in the charge carriers (holes). The inverse occurs when capturing negative charges raises because conductivity by the holes increases. The application of this principle allows improved detection by coating nanomaterials such as carbon nanotubes, graphene, and MNPs on the electrodes and the sensing platforms [91]. In this way, biosensors can be designed for various pharmaceutical and therapeutic drug-related work. Figure 7 below shows a FET sensor used in the detection of glucose oxidase using a graphene biosensor.

In Figure 7, graphene is used as a high electrical conducting platform which changes the current on both the source and the drain electrodes when a substance adsorbs on its surface. In this case, glucose oxidase enzyme (GOD) is used together with a crosslinker molecule to detect changes in conductance when glucose binds to the enzyme [92]. Other applications of FET sensors for various pharmaceutical agents have been published. Zinc oxide nanoribbons were used as transducer materials for the detection of glucose in phosphate buffer solution (PBS) and a LOD of 70 µM and linear range of 0–80 mM were reported [93]. In another study, insulin was analyzed in PBS on graphene transducers with detection limits at 35 pM [94]. Although FET devices offer high sensitivity, selectivity, miniaturization, and low power use as advantages, dielectric membranes are sensitive to motion, which affects the accuracy of detection. Lastly, because FET uses biomarkers, it is not regarded as a label-free method, which affects the applicability of this technique due to cost implications [95].

#### 3.1.2. Electrochemical Detection Methods

Electrochemical detection involves the use of functionalized electrodes for the detection of target analytes and signal transduction. Like FET devices, electrochemical methods translate binding affinities into readable electrical signals. The difference between the two methods is that electrochemical techniques require the use of an electrolyte solution as a conducting medium instead of solid-state transducers found on FET devices [96]. Figure 8 below shows a general schematic of an electrochemical system.

Figure 8 above shows how an electrochemical sensor is prepared and used in detection experiments. From position (a), a glass-based carbon (GCE) electrode was functionalized with gold nanoparticles followed by thioglycolic acid and lectins (b,c), while simultaneously blocking nonspecific adsorption to the electrode using BSA (d). Lectin–au-thionine bioconjugates were adsorbed on the electrode as recognition elements (e). Finally, the electrochemical apparatus is assembled and utilized to produce a signal of intensity (amperes) versus electric volts (f) [97]. Systems like these have been designed for a wide range of applications for pharmaceutical and therapeutic targets. For example, studies have shown the detection of dopamine using graphene functionalized with PVP as the sensing platform on the GCE. The study reported a LOD of 0.2 nM and a linear range up to 10^−^^10^ with an r^2^ of 0.99 [98]. Research on cholesterol was also undertaken using chitosan, graphene hybrid nanocomposites, reaching a LOD of 0.75 µM and a linear range of 2.2–520 µM [99]. Furthermore, extensive research into antineoplastic drugs using electrochemical sensing has been explored. Flutamide detection by silver coated GCE produced LOD (mol L^−^^1^) in the 10^−^^6^ and a linear range of 1–100 × 10^−^^5^ (mol L^−^^1^) [100]. In a similar study, gemcitabine was detected and analyzed using amino thiophenol-functionalized gold nanoparticles where LOD in the 10^−^^15^ and a linear range of 10^−^^8^ to 10^−^^15^ (mol L^−^^1^) [101]. The use of electrochemical sensing is clearly a prominent field of study, and its success can only bring more advantages for chemical sensing related to a variety of pharmaceutical drugs. Although the LOD and linear range of this method are extremely sensitive, the short lifespan is still a limitation and as such [102], other techniques have been approached to solve this issue.

### 3.2. Optical Detection Methods

Light/matter interaction principles have been widely explored for the investigation of organic and inorganic molecules. As such, a lot of research has been channeled towards using optical methods for detecting and collecting information on analytes adsorbed on nanomaterial scaffoldings or sensors. Nanomaterials carry the property of surface plasmons which aid in the detection method of the techniques [103]. Thus, in this section, we discuss popular plasmon-related methods used in pharmaceutical drug applications.

#### 3.2.1. Surface Plasmon Resonance Spectroscopy

Surface plasmons found on the surface of a nanomaterial can be used for trace detection and monitoring of small molecules using changes in the refractive index as a signal in real time. In principle, when light photons approach a nanomaterial such as gold NPs at an angle, a refractive index can be obtained from the plasmon wave and set as a calibration point. When the MNP is functionalized with polymers, recognition elements, and analytes, changes in the refractive index are then recorded again and compared. Since the refractive index is related to the surface of the substrate, changes in the refractive index can also be correlated to binding affinities and used in medical diagnostics, drug detection, and virus monitoring among other applications [104]. Figure one below shows the principle of SPR and how it is used in biomedical applications.

Figure 9 shows the principle of SPR used in protein detection applications. From left to right, a glass slide is seen coated with layers of gold (Au) and a variety of antibodies. In the second step, anti-PSA is used as biorecognition elements for the detection of PSA. At each step, laser light is directed to the biosensors and changes in the refractive index in relation to the binding of molecules [105]. Examples of SPR applications can be seen in a wide range of biomedical and pharmaceutical experiments. Studies have shown the diagnosis of malignant and infectious diseases using the biomarker rhodamine 6G and SPR detection [106]. In industrial applications, aflatoxin, a toxic small molecule found in dairy products, was detected at LOD values of 76 pM using a silicon photonic biosensor [107]. Smartphone platforms are also being incorporated to SPR detection where 50 nm gold nanofilms are used for sensing immunoglobin G (IgG) producing LOD values from 15–47 nM [108]. Silver nanospheres and nanorods on titanium oxide substrates were used for the detection of streptavidin obtaining a LOD value of 0.3 µg/mL using halogen lamp technology [109]. Bromocriptine, a pharmaceutical drug used to treat menstrual problems, was investigated using laccase immobilized on a carboxymethyl dextran functionalized SPR sensor, where a detection limit of 10^−^^2^ ng/mL to 10^3^ ng/mL was reported [110]. The values reported above show that the common feature of the SPR technique is high sensitivity and rapid analysis. However, SPR has limitations such as long lag times, sensitivity to temperature and motion as well as continuous optical alignment and maintenance [111].

#### 3.2.2. Surface Enhanced Raman Spectroscopy (SERS)

Amongst non-destructive methods of detection, Raman spectroscopy has become a favorite method for qualitative experiments because it allows molecular fingerprinting of organic and organometallic substances. Light–matter interaction produces scattered radiation which is collected at right angles to the surface. The scattered radiation produces different frequencies from the laser wavelength due to inelastic scattering caused by photon–molecule interactions. Raman shifts expressed in wavenumbers can be correlated to bond fragments of analytes as means of detection. A major limitation of this method is low signal output which is normally 0.01% of the radiation. To solve this issue, nanomaterials were incorporated into the technique by employing surface plasmonic resonance for signal enhancement purposes. Such as, surface-enhanced Raman spectroscopy (SERS) has brought renewed interest and trust in SERS as a pharmaceutical drug detection method [112]. Figure 10 below illustrates both the fabrication of SERS scaffolds and their application in qualitative analysis.

In the above image, a 632.8 nm laser source is exciting a set of carbon-based scaffolds of various dimensions and functionality. Gold quantum dots (GQDs, 0D), carbon nanotubes (CNTs, 1D), graphene oxide (GO, 2D) and reduced graphene oxide (rGO, 2D), and gold nanohybrids (GHs, 3D) were further functionalized with 4-mercaptobenzoic acid (4-MBA, 10^−^^6^ M) as the analyte. From observation of the Raman spectra, it is seen that vibrational modes are associated with 4-MBA. The spectra also show increased signal intensity amongst the scaffolds, with GQDs providing the highest intensity. Specifically, the study reports surface enhancement factors of 10^7^ from GQDs followed by 10^6^ from GO-Au NPs, which is a significant improvement on the Raman signals obtained, making SERS a reliable technique for small molecule detection. More examples like the one above have been reported in the literature. Levofloxacin (antibiotic) was detected from artificial urine using hydroxylamine silver nanoparticle microfluidic devices and SERS where the quantitative analysis yielded a LOD of 0.07 mM [114]. A similar pharmaceutical study quantified promethazine using the same microfluidic device and detected concentrations as lows 10^−^^7^ M [115]. Medication for the treatment of hypertension known as captopril was obtained from human blood and analyzed using SERS and citrate functionalized silver nanoparticles. It was reported that a LOD of 0,4 µM was achieved from the quantitative calculations [116]. Lastly, the common over-the-counter medication aspirin, was analyzed on silver nanoparticles supported by filter paper, the linear range reported in this study spanned from 0.1 to 1 mM [68]. The examples shown indicate that SERS is a highly versatile, sensitive, and non-destructive detection method; however, issues such as fluorescence noise, long acquisition time, and unstable lasers still hamper the full potential of this method [117]. With further research and optimization, SERS systems will surely improve knowledge on pharmaceutical drug design and quality control, which is essential for healthcare reasons.

## 4. Prospects and Shortcomings

The pharmaceutical industry is one of the pillars of our health systems around the world. Research and knowledge dissemination in this field is important for our global health status. This review provided a brief overview of technologies used in the detection and analysis of therapeutic drugs that are used to treat serious diseases such as cancer, hypertension, viruses, and tumors. Nanotechnology has made great strides in providing sensor applications for pharmaceutical drug design and monitoring. Properties of polymeric, metallic, and carbon-based substrates can be fabricated into sensor platforms that allow the adsorption of analytes for analysis purposes. Furthermore, electrical and optical detection methods coupled with nanomaterials have enhanced sensor application research which is a positive for the pharmaceutical industry. Parameters such as limit of detection, linear range, and sensitivity show improved responses towards analytes being investigated. With future research and design, challenges related to toxicity, shelf life, and API quality can be overcome by improved drug design, monitoring, and targeted sensing. The major limitations observed from this study start with the challenges in reproducing sensing platforms consistently using less-expensive methods. Secondly, electrical-based detection methods are sensitive to motion, and they struggle to detect multiply layered sensors, which makes calibration and label-free applications a tedious process. Thirdly, optical methods suffer from long acquisition times, laser instability, continuous alignment maintenance, and fluorescence noise, which requires optimization steps prior to acquisition. Technical challenges aside, nanomaterial-based research into pharmaceutical drugs is a very important field of science and its understanding can only be improved by more research, innovation, and application.

## Figures and Tables

**Figure 1 nanomaterials-12-02688-f001:**
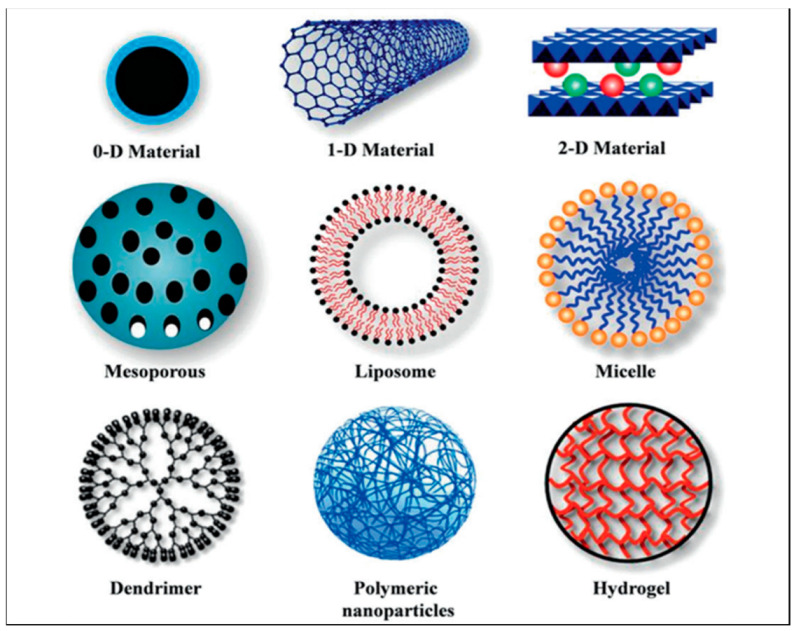
Classes of nanoparticles. Each subclass consists of unique properties for drug delivery research. (Reproduced from Ref. [33] with permission from Nature).

**Figure 2 nanomaterials-12-02688-f002:**
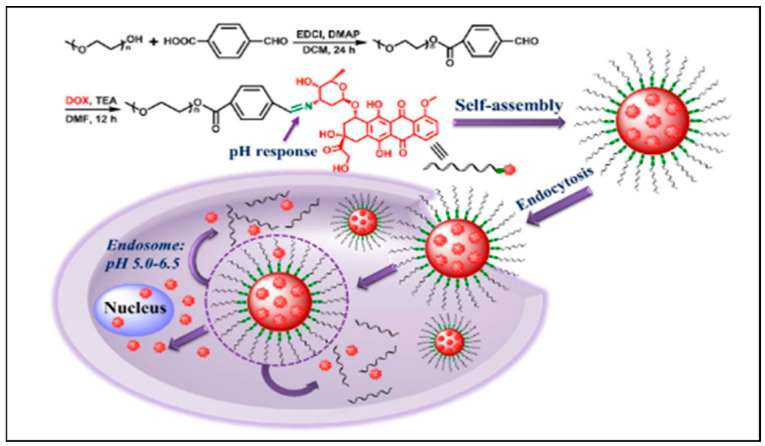
Illustration of self-assembled polymer PEG-Schiff nanoparticles and DOX drug delivery using pH sensitive conjugate PEG-Schiff-DOX. (Reproduced from Ref. [45] with permission from Frontiers).

**Figure 3 nanomaterials-12-02688-f003:**
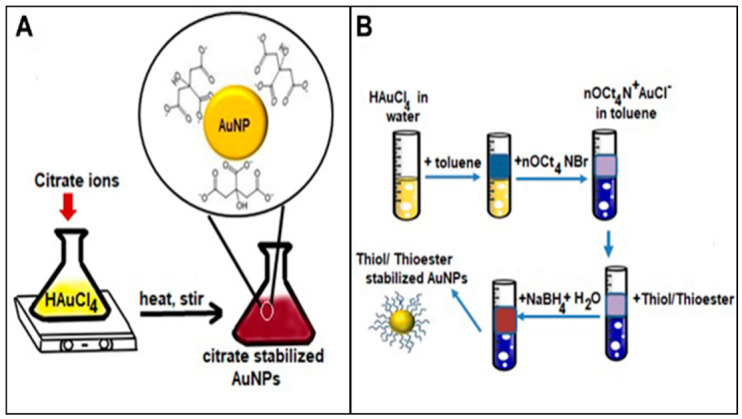
Synthesis of functionalized gold nanoparticles. (**A**) Turkevich method, citrate stabilized. (**B**) Burst method, thioester stabilized. (Reproduced from Ref. [13] with permission from author).

**Figure 4 nanomaterials-12-02688-f004:**
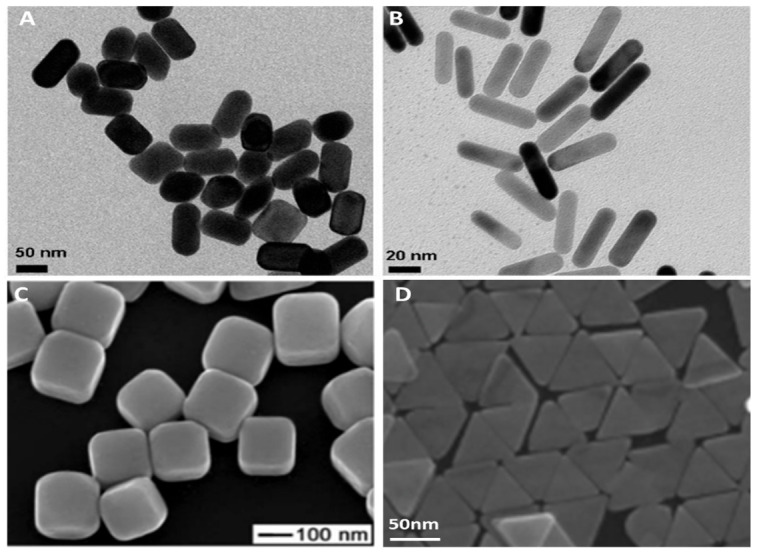
Images of different shapes of nanoparticles used in drug detection. (**A**,**B**) TEM images of gold nanorods. (**C**) SEM image of silver nanocubes. (**D**) SEM image of triangular nanoplates (Reproduced from Ref. [69] with permission from IOP.) (Reproduced from Ref. [70] with permission from RSC).

**Figure 5 nanomaterials-12-02688-f005:**
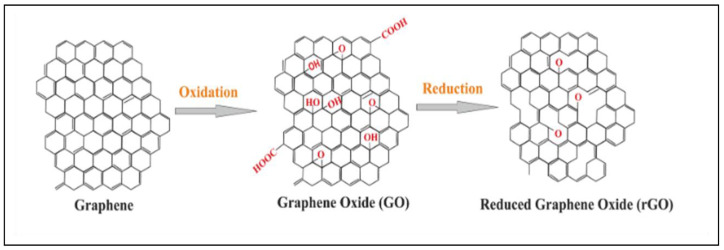
Chemical conversion of graphene-to-graphene oxide (GO) and reduced graphene oxide (rGO). (Reproduced from Ref. [83] with permission from RSC).

**Figure 6 nanomaterials-12-02688-f006:**
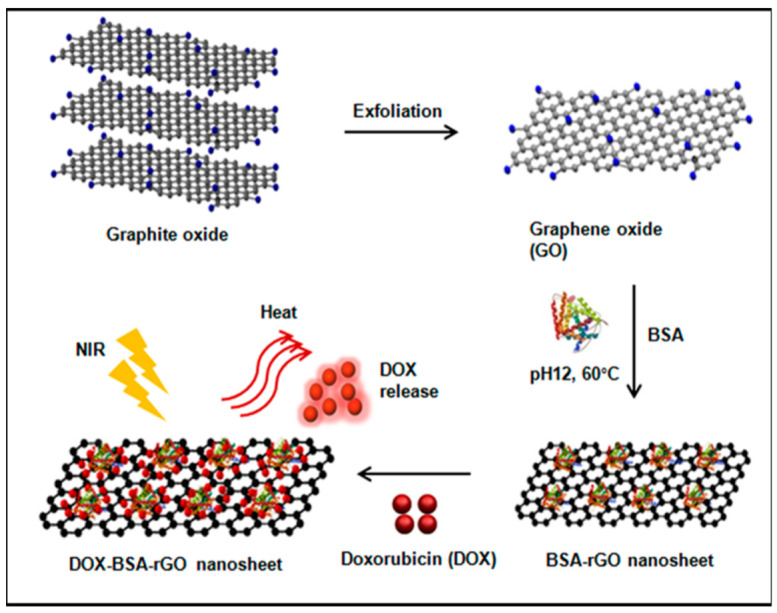
Synthesis and application of rGO as a DOX drug carrier. (Reproduced from Ref. [86] with permission from American Chemical Society. Copywrite 2016).

**Figure 7 nanomaterials-12-02688-f007:**
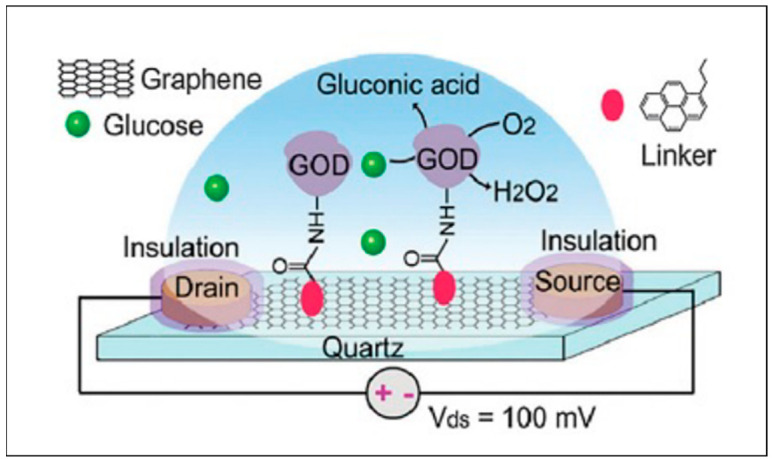
Illustration of a FET system for glucose detection using a graphene-based platform. (Reproduced from Ref. [92] with permission from RSC).

**Figure 8 nanomaterials-12-02688-f008:**
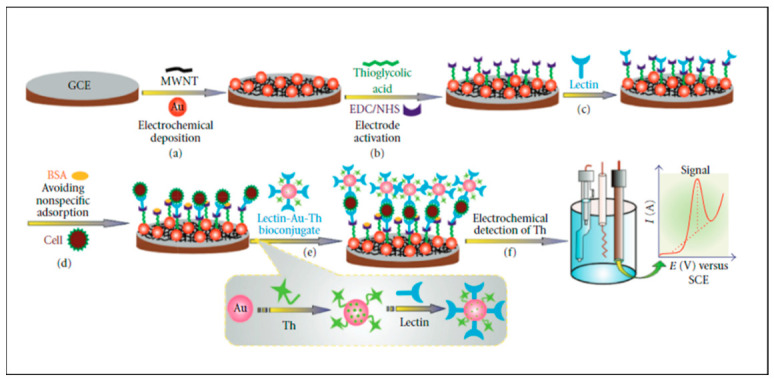
Schematic of synthesis and application of a glycan electrochemical sensor. (Reproduced from Ref. [97] with permission from Hindawi).

**Figure 9 nanomaterials-12-02688-f009:**
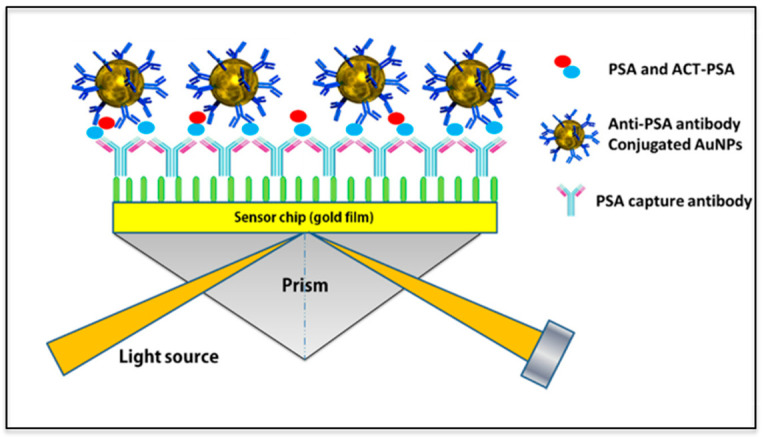
Schematic of SPR configuration used in the detection of PSA. (Reproduced from Ref. [105] with permission from MDPI).

**Figure 10 nanomaterials-12-02688-f010:**
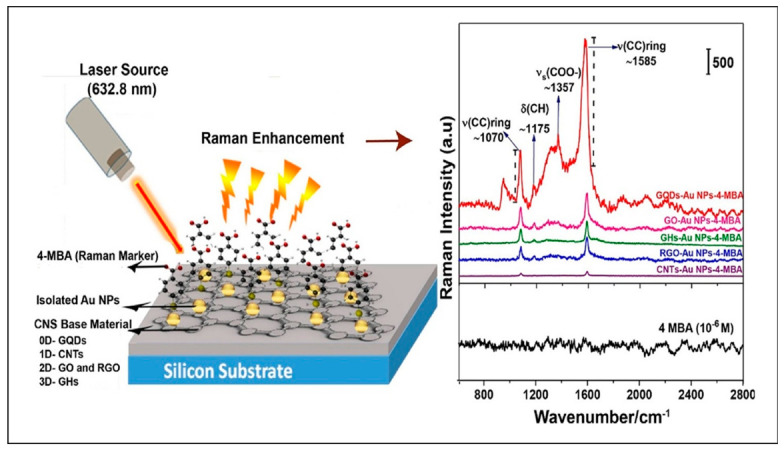
Schematic of a carbon-based SERS substrate of various functional groups. (**left**). SERS spectra of the analyte 4-mercaptobenzoic acid (MBA) adsorbed on the scaffolds (**right**). (Reproduced from Ref. [113] with permission from Springer).

## Data Availability

Data are available via personal communication with proper reasons.

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
