# Peer review of "Chemical Sensor Nanotechnology in Pharmaceutical Drug Research"

_nanomaterials, 2022, doi:10.3390/nano12152688_

Round 1

Reviewer 1 Report

The authors present in this review an overview of chemical sensors for drugs detection analysing different approaches. It is a topic of great interest, considering the future demand for health sensors. The different sections are well organised and text is clear.

Therefore, I consider that the manuscript is suitable for publication in this journal

Reviewer 2 Report

The present review reports recent research on chemical sensor nanotechnology for applications in drug research.

The topic is highly interesting especially in the current situation. 

The review is clearly written and well organized, so I suggest its publication after minor revisions.

In Figure 1, the picture of Polymeric nanostructures is difficult to understand and very similar to metal nanostructures. I suggest the authors to better explain and draw their possible structure.

In addition, please check the caption of Figure 1: is Ref44 correct? I couldn’t find the picture in such paper.

The first sentence of section 3.2 is not clear, I suggest the authors to clarify its meaning.

Finally, I suggest to check the correspondence between figures’ numbers and figures’ captions along the text, and to clearly indicate the figure numbers in the text.

Reviewer 3 Report

The Review entitled: “Chemical sensor nanotechnology in pharmaceutical drug research”, in view of the recent pandemic situation, provides a detailed overview of information about the development of quality control methods employed in pharmaceutical and health research.

Despite the manuscript is well organized and it reports the main case studies from the literature, some grammatical errors and punctuation required minor revisions before the publication.

Furthermore, in order to improve the quality of the work, a further investigation of the most recent literature related the future perspective in the research field of quality control in drug manufacturing aimed at overcoming the limits of quality control techniques studied until now, it is suggested.
